# Anti-Bacterial Properties and Biocompatibility of Novel SiC Coating for Dental Ceramic

**DOI:** 10.3390/jfb11020033

**Published:** 2020-05-20

**Authors:** Samira Esteves Afonso Camargo, Azeem S. Mohiuddeen, Chaker Fares, Jessica L. Partain, Patrick H. Carey, Fan Ren, Shu-Min Hsu, Arthur E. Clark, Josephine F. Esquivel-Upshaw

**Affiliations:** 1Department of Restorative Dental Sciences, Division of Prosthodontics, University of Florida College of Dentistry, Gainesville, FL 32610, USA; scamargo@dental.ufl.edu (S.E.A.C.); AMohiuddeen@dental.ufl.edu (A.S.M.); shuminhsu@ufl.edu (S.-M.H.); BCLARK@dental.ufl.edu (A.E.C.); 2Department of Chemical Engineering, University of Florida Herbert Wertheim College of Engineering, Gainesville, FL 32611, USA; c.fares@ufl.edu (C.F.); jpartain3@ufl.edu (J.L.P.); careyph@ufl.edu (P.H.C.IV); fren@che.ufl.edu (F.R.)

**Keywords:** antimicrobial, ceramic, coating, silicon carbide, cytotoxicity

## Abstract

A 200 nm plasma-enhanced chemical vapor-deposited SiC was used as a coating on dental ceramics to improve anti-bacterial properties for the applications of dental prosthesis. A thin SiO_2_ (20 nm) in the same system was deposited first, prior to SiC deposition, to improve the adhesion between SiC to dental ceramic. Silane and methane were the precursors for SiC deposition, and the SiO_2_ deposition employed silane and nitrous oxide as the precursors. SiC antimicrobial activity was evaluated on the proliferation of biofilm, *Streptococcus sanguinis*, and *Streptococcus mutans* on SiC-coated and uncoated dental ceramics for 24 h. The ceramic coating with SiC exhibited a biofilm coverage of 16.9%, whereas uncoated samples demonstrated a significantly higher biofilm coverage of 91.8%, measured with fluorescence and scanning electron microscopic images. The cytotoxicity of the SiC coating was evaluated using human periodontal ligament fibroblasts (HPdLF) by CellTiter-BlueCell viability assay. After 24 h of HPdLF cultivation, no obvious cytotoxicity was observed on the SiC coating and control group; both sets of samples exhibited similar cell adhesion and proliferation. SiC coating on a ceramic demonstrated antimicrobial activity without inducing cytotoxic effects.

## 1. Introduction

Improving the dental restoration’s lifetime through careful selection of materials has been the focus of many previous studies, where traditionally different alloys and metal systems have been evaluated for biocompatibility and antibacterial properties [1,2,3,4,5,6,7,8,9,10]. Although the majority of these approaches have used methods involving the bulk system, use of thin films can offer an easier avenue to maintaining the integrity of the restoration while grafting unique properties to selective areas where the film is applied. One of the novel thin films of interest is silicon carbide due to the ease of deposition, lack of reactivity to the oral environment, and high strength [11,12,13,14,15] of this material. Silicon carbide has found significant success in two major areas: semiconductors and ceramics. Silicon carbide can easily be deposited by plasma deposition systems to form conformal films with tight control over thickness from nanometers to micrometers [16,17,18,19,20]. Silicon carbide may present a biosafe path to protect restorative surfaces from bacterial adhesion and degradation without compromising the bulk properties of traditional dental material technology.

In recent reports, the development and optimization of SiC-based coatings to minimize surface corrosion and wear on glass ceramic veneers has been demonstrated [14,21]. Hsu et al. [21] reported that SiC-coated dental ceramics demonstrated a significant reduction in corrosion when placed in caustic environments that represent the extreme conditions of the oral cavity. Chen et al. [22] demonstrated that the thickness of the SiC coating can also be optimized to match any tooth shade required within the dental field. 

As new dental materials such as SiC are developed, analyzing their anti-bacterial and biocompatibility properties is imperative. The biofilms that adhere to dental materials can be described as a micro-ecosystem formed by various species of microorganisms, surrounded by a protein extracellular matrix and polysaccharides generated by them [23]. These biofilms are responsible for plaque formation, leading to dental caries, periodontal disease, peri-implantitis, and enamel demineralization. They can adhere to both biotic and abiotic surfaces, such as prostheses, implants, or host tissues [24]. Bacterial adhesion to a substrate is a multifactorial process that involves surface properties inherent to both the bacteria and the biomaterial [25,26]. Bacteria present in the oral cavity naturally tend to adhere to ceramic materials and also to the interface between tooth and restoration [27,28]. They adhere on the cervical third of the proximal surface, and along the gingival margin where they are protected from mechanical action, creating a well-structured biofilm [29].

Understanding how these biofilms adhere to surfaces is crucial in qualifying new dental materials. On solid surfaces such as enamel, the ability to aggregate, the order of appearance of the microorganisms [30,31], and the environment [23] are essential factors in oral biofilm formation. The composition of the material, as well as the surface structure, can influence the initial bacterial adhesion and compromise dental health. Poor oral hygiene or a compromised immune system can also be contributing factors that aid colonization [32]. SiC coating provides a smoother surface after corrosion, which could minimize plaque accumulation, secondary caries, and periodontal inflammation from occurring [21].

Although SiC has been studied and applied as a viable biomaterial in the literature [15,16,17,18], no work to date has studied the feasibility of SiC as a dental material. One of the precluding tasks to determine if SiC can be utilized as a dental material is to quantify how the coating affects monomicrobial and polymicrobial biofilm adhesion, as well as the cytotoxicity for human cells.

The objective of this study was to evaluate the effect of SiC coating on monomicrobial and polymicrobial biofilm adhesion, as well as determine the viability of human cells in contact with the coating surface. Understanding the biological and mechanical behavior of dental materials in the oral environment is crucial to developing predictable restorations, and thus we proposed the hypotheses that SiC coatings on ceramic will (1) decrease monomicrobial and polymicrobial biofilm adhesion on the surface of the ceramic, and (2) exhibit biocompatibility without signs of cytotoxicity. 

## 2. Material and Methods

### 2.1. Ceramic Sample Preparation 

Fluorapatite glass-ceramic disks (Ivoclar Vivadent AG, Schaan, Liechtenstein, 12.6 × 2 ± 0.2 mm) were polished using silicon carbide abrasive paper (Carbimet, Buehler, Lake Bluff, IL, USA) and subsequently cleaned using the following procedure: (1) acetone cleaning in an ultrasonic bath, (2) isopropyl alcohol rinse followed by a compressed nitrogen drying step, and (3) ozone treatment to remove surface carbon contamination. 

### 2.2. SiC Coating 

After the ceramic disks were polished and cleaned, half of the disks (40/80) were coated with silicon dioxide (SiO_2_) and silicon carbide (SiC) on both sides. These depositions were done within a plasma-enhanced chemical vapor deposition system (PECVD, PlasmaTherm 790, Saint Petersburg, FL, USA). The corresponding thicknesses of the deposited SiO_2_ and SiC films were 20 nm and 200 nm, respectively. The SiO_2_ film was utilized as an adhesion layer between the ceramic and SiC, and was optimized in a previous report [21]. The precursors for the SiO_2_ film were silane (SiH_4_) and nitrous oxide (N_2_O). Following the SiO_2_ deposition, methane (CH_4_) and silane (SiH_4_ ) were the precursors used for silicon carbide deposition. The deposition temperature for both films was 300 °C, and the deposition rates for SiO_2_ was 330 Å/min and for SiC was 170 Å/min [14]. 

### 2.3. Bacteria Growth

To study the antimicrobial properties of the coated and uncoated disks, monomicrobial and polymicrobial strains (ATCC—American Type Culture Collection) of *Streptococcus mutans* (ATCC 35688) and *Streptococcus sanguinis* (ATCC BAA-1455) were utilized. *S. mutans* and *S. sanguinis* were used for this study because these bacteria are the early colonizers of initial supragingival biofilm in the first 8 h and are present in greater quantities in the oral biofilm. The strains were grown onto agar plates with brain heart infusion broth (BHI, Himedia) for 24 h at 37 °C. After the growth, each microbial suspension was centrifuged at 4700 rpm for 10 min (Centrifuge model MPW-350, Beckman Coulter, Indianapolis, IN, USA) to separate the supernatant and microbial suspension. The centrifuge process was performed twice to minimize the quantity of debris. After separation, the microbial suspension adjusted to 10^7^ colony-forming units (CFU)/mL and was ready to be placed onto the coated and uncoated glass-ceramic disks.

### 2.4. Experimental Design

The two groups in this study were (i) non-coated fluorapatite glass-ceramic disks as the reference group, and (ii) SiO_2_/SiC-coated fluorapatite glass-ceramic disks (SiC-disks). Twenty-four ceramic specimens of each group were sterilized in an autoclave (121 °C, 60 min) and then placed into individual sterile plates. A total of 12 samples from each group were used to study the effects of monomicrobial suspensions, and the remaining 12 samples were used to study polymicrobial suspensions (*S. sanguinis* and *S. mutans*). For all samples, 100 uL of mono- or polymicrobial suspension was added to each plate containing a coated or uncoated disk, along with 1 mL of BHI broth to adequately cover the full disk. 

To study the biocompatibility and cytotoxicity of the SiC coating, human periodontal ligament fibroblasts (HPdLF, Lonza, Basel, Switzerland) were cultured and subsequently placed onto coated and uncoated samples. The HPdLF were cultured at 37 °C in a growth media consisting of Dulbecco’s modified Eagle’s medium (DMEM) with 10% fetal bovine serum and 1% penicillin/streptomycin. After the cells were cultured, they were seeded onto 16 sterilized ceramic disks from each group, each within its own sterilized well plate. The approximate concentration of cells after seeding was 20,000 cells per well.

The quantitative data were presented as the means ± standard deviations. The statistical differences were compared using one-way ANOVA and Tukey’s test (Graph Prism 6.0, GraphPad Software Inc., San Diego, CA, USA). For all analyses, statistical significance was pre-set at α = 0.05.

### 2.5. Characterization Techniques 

#### 2.5.1. Scanning Electron Microscopy

Non-coated, SiC-coated, and glazed disks were examined under scanning electron microscopy using a MAICE system (JEOL JSM-6400 Scanning Electron Microscope, JEOL LTD, Tokyo, Japan) to characterize surface roughness. After the coated and non-coated disks were incubated for 24 h, the culture medium was removed, and the polymicrobial biofilm adhered to the samples was fixed in a solution of 3% glutaraldehyde, 0.1 mol/L sodium cacodylate, and 0.1 mol/L sucrose for 45 min. Samples were soaked for 10 min in a buffer solution of 0.1 mol/L sodium cacodylate and 0.1 mol/L sucrose. Sample surfaces and cells were processed in serial ethanol dehydrations for 10 min each and dehydrated in hexamethyldisilazane (HDMS) before being stored in a desiccator until SEM imaging. The samples were then sputter-coated with a palladium–gold alloy (Polaron SC 7620 Sputter Coater, Quorum Technologies, Laughton, East Sussex, United Kingdom) with a thickness of 10 nm to reduce charging effects during SEM analysis (10–15 mA, under a vacuum of 130 mTorr). After this, the SEM was operated at 5 kV, spot 3 to 6 (FEI NOVA 430, Hillsboro, OR, USA).

#### 2.5.2. Atomic Force Microscopy (AFM)

Topographies of non-coated and coated with SiC samples were evaluated using an atomic force microscopy system (Bruker/Veeco/Digital Instruments NanoScope V, Billerica, MA, USA). The AFM was operated in tapping mode, using a silicon AFM probe (RTESP-300, Bruker, Billerica, MA, USA), with radius less than 10 nm, and resonance frequency between 200 and 400 kHz. 

#### 2.5.3. Water Contact Angle Measurements

The contact angle is described as the angle between the liquid–solid interface. The water contact angle (WCA) measurements were performed on the neutralized non-coated and coated SiC surfaces (neutralized up to pH 6.0). Static contact angles were assessed by the sessile drop method with a contact angle goniometer (Krüss DSA 10, Matthews, NC, USA), equipped with video capture. The automatic dosing feature of the DSA 10 dispensed a water drop on the non-coated and coated SiC surfaces, and the needle was manually withdrawn. Images were captured after contact of a droplet with the surface by a camera leveled with the surface. Contact angle measurements were analyzed by the circle fitting profile available with the DSA 10 imaging software. Three separate measurements were made on each sample at different locations.

#### 2.5.4. Cytotoxicity Test

Cell viability were determined by a CellTiter-BlueCell Viability Assay (Promega G808A), which was used according to the manufacturer’s instructions. After 24 h, 50 μL of CellTiter-Blue dye was added to ceramic specimens for every 500 μL of culture media, and samples were incubated for 6 h at 37 °C and 5% CO_2_. Sample fluorescence was analyzed using a spectrophotometer (SmartSpec Plus, Bio-Rad, Hercules, CA, USA) at a wavelength of 600 nm, which generated density optic values. 

#### 2.5.5. Colony-Forming Units 

After 24 h of incubation, the ceramic samples were removed and placed inside an Eppendorf tube with Ringer’s solution (Sigma-Aldrich). The monomicrobial biofilm of *S. sanguinis* and *S. mutans* was disaggregated by an ultrasonic homogenizer (Sonopuls HD 2200–Bandelin Electronic, Berlin, Germany) for 30 s. The generated microbial suspension was serially diluted (1:10), and 100 μL was seeded on BHI agar for each respective sample. After 48 h of incubation, the concentration of colony-forming units (CFU) per milliliter was determined by visual counting.

#### 2.5.6. Fluorescence Assay 

After the incubation period, bacteria adhered to the samples were stained with SYTO 9 dye (Live/Dead BacLight Bacterial Viability Kit, ThermoFisher Scientific, Waltham, MA, USA) according to the manufacturer’s instructions. Fluorescence images of the live bacteria were recorded in a fluorescence microscope (Zeiss Imager-A2, Jena, Germany) and analyzed by ImageJ software. Bacteria coverage percentages were averaged over five random areas on each filter specimen (*n* = 8).

The quantitative data are presented as the means ± standard deviations. The statistical differences were compared using one-way ANOVA and Tukey’s test (Graph Prism 6.0, GraphPad Software Inc., San Diego, CA, USA). For all analyses, statistical significance was pre-set at α = 0.05.

## 3. Results

### 3.1. Atomic Force Microscopy (AFM) and Scanning Electron Microscopy (SEM)

According to Figure 1, the AFM analysis showed an irregular surface topography in non-coated samples with roughness Rq 9.58. In contrast, SiC-coated samples were slightly smoother with Rq 7.98 (*p* > 0.05). Non-coated, SiC-coated, and glazed disks demonstrated significant qualitative differences in surface roughness (Figure 2A–F).

### 3.2. Water Contact Angle Measurements (WCAs)

The outcomes indicate different WCAs between the non-coated and coated surfaces (Table 1). The coated SiC surface presented higher contact angles than the non-coated surface (*p* > 0.05). 

### 3.3. Bacterial Growth

After 24 h, the amount of polymicrobial biofilm (*S. mutans* and *S. sanguinis*) was less on the SiC-coated disks. The biofilm coverage on these disks was 16.9%, whereas uncoated reference samples showed a significantly higher biofilm coverage of 91.8% (*p* < 0.0001) (Figure 3). 

The fluorescence images in Figure 4 demonstrate higher polymicrobial biofilm (*S. mutans* and *S. sanguinis*) formation on the control group than on the coated group. The SEM images confirmed the results from the live assays, showing a reduction in the number of adherent biofilms on the coated group for *S. mutans* and *S. sanguinis* after 24 h of culture (Figure 5).

There was a significant reduction in the number of CFU/mL after 24 h contact with ceramic SiC coating compared to the control group (Figure 6), showing *p* = 0.0003 for *S. sanguinis* and *p* ≤ 0.0001 for *S. mutans*. The reduction of the monomicrobial biofilm of *S. sanguinis* and *S. mutans* were also similar (*p* = 0.2528).

### 3.4. Biocompatibility Testing

In order to determine whether the SiC coating presented biocompatibility, the cytotoxicity was evaluated by the CellTiter-Blue assay. After 24 h of HPdLF cultivation on the samples, no obvious cytotoxicity was observed, as evidenced by the absorbance of the cells cultured on SiC coating being comparable to the control group (*p* = 0.3904) (Figure 7). 

The SEM images showed the interaction of cell extensions with the SiC-coated and non-coated surfaces. SEM results demonstrated that cells adhered and covered the full surface of the disk after 24 h in culture. Additionally, the cell morphology was similar for cells cultured on SiC-coated and non-coated samples, where HPdLFs appeared oval-shaped and flattened on the surface (Figure 8).

## 4. Discussion

The adhesion of streptococci to solid surfaces has been described as a two-step process, where the first step is influenced predominantly by macroscopic substratum properties such as surface roughness and surface free energy [33]. High substratum surface roughness has been shown to induce increased plaque formation in vivo [34,35], as the initial adhesion of microorganisms preferably starts at locations that provide shelter against shear forces. The surface roughness of differently prepared lithia disilicate ceramic restorations (pressed (Press), milled (CAD), fluorapatite layered (ZirPress/Ceram), and glazed (Ceram Glaze)) is closely related to the adherence of *S. mutans* [27]. The application of a SiC coating on the surfaces of ceramic significantly minimized biofilm adhesion on the surface. A possible explanation is that SiC altered the surface roughness of the ceramic. SEM 3D analysis of ceramic surfaces before and after coating with SiC (Figure 2A–F) revealed a distinct difference in surface roughness between the two surfaces. The SiC coating achieved planarization of the ceramic surface, essentially minimizing surface roughness that bacteria can adhere to. Even when compared to a glazed ceramic surface, the SiC coating produced a smoother surface (Figure 2B,C). This change in surface roughness was further confirmed by a difference in surface topography of coated and non-coated surfaces, as observed by AFM (Figure 1).

Numerous studies have also demonstrated that surfaces with high surface free energy foster microbial adherence in vitro and in vivo [35,36,37,38]. In addition, bacteria with high cell surface free energy appear to adhere preferentially to surfaces with high surface free energy. 

Bacterial adhesion to a substrate and the initial biofilm composition is also related to surface hydrophobicity, and communication between existing microorganisms [28,37,38]. Although the literature shows that bacteria adhere better to hydrophobic surfaces, biofilm formation depends not only on the surface topography but also on the bacteria species involved. Evidence suggests that the presence of polysaccharides on the cell surface of Gram-positive bacteria, such as *S. mutans* and *S. sanguinis*, tends to make the bacterial cell more hydrophilic [35,38]. Additionally, the cell shape and size, including surface characteristics such as extracellular substances such as flagella and fimbriae, as well as the production of proteins, are believed to have a significant influence on the process of bacterial adhesion [33,34].

On the basis of contact angle measurements conducted in this study, we found that coated SiC surfaces were more hydrophobic than non-coated surfaces; however, we found lower colonized bacteria. This might suggest that SiC-coated samples can cause the detachment of bacteria from the biofilm after 24 h.

In vitro studies have been performed on bacterial adhesion to various ceramic materials [28,39,40,41,42,43]. In the present study, we verified the adhesion of streptococcal species and the accumulation of complex biofilms, which are of critical importance to predict the effect of biofilms on the surface ceramics. *Streptococcus mutans* and *Streptococcus sanguinis* were used for this study because these bacteria are predominant members in dental plaque, which is often described as a dual-species biofilm model [36,44]. The association of these microorganisms can interfere in the development of each other and promote different results between polymicrobial and monomicrobial biofilms.

Lower adhesion of polymicrobial biofilm of *S. mutans* and *S. sanguinis* on the surface of fluorapatite glass-ceramic coated with SiC was demonstrated. Additionally, we found a significant reduction in their monomicrobial biofilm. Two studies found significant differences in adhesion of different streptococcal species among different ceramic materials. These differences were observed in the bacterial surface coverage, as well as in the thickness of the biofilm between the tested ceramic materials. The lowest surface coverage (19.0%) and biofilm thickness (1.9 mm) were determined on the HIP Y-TZP ceramic; the highest mean values were identified with the lithium disilicate glass-ceramic (46.8%, 12.6 mm) [41,42]. On the other hand, Meier et al. [40] did not observe any significance. Our study showed that the percentage of polymicrobial biofilm coverage was five times less for the ceramic coated with SiC (16.9%) compared with the uncoated group (91.8%). Ceramic coating also presented significant reductions of CFU/mL in monomicrobial biofilms of *S. mutans* and *S. sanguinis*. This may be related to the alteration of the surface energy and slightly smoother surface of coated SiC surfaces, which can decrease the bacteria adhesion. 

Because significant differences in bacteria adhesion were identified among the groups, the hypothesis that SiC coatings on ceramic will decrease monomicrobial and polymicrobial biofilm adhesion on the surface of the ceramic has to be accepted. 

Furthermore, a study by Auschill et al. [45] found that biofilms form a very thin layer on ceramic, but demonstrated the highest viability compared with biofilms on amalgam, gold, and composite resin. Dal Piva et al. [28] found a lower value of bacterial adhesion for zirconia reinforced lithium silicate (ZLS) ceramic compared to zirconia partially stabilized by yttrium (YZHT) ceramic. Lithium disilicate glass-ceramics (LDS) have lower *S. mutans* adhesion than fully stabilized zirconia (FSZ) and partially stabilized zirconia (PSZ), but the difference was not reflected in early biofilm formation [46]. Polished feldspathic ceramics showed less *Candida albicans* adherence, whereas the adherence of streptococci was greater than *C. albicans* in all conditions [39].

Ceramic materials may exhibit positive or negative results in terms of cell viability, according to the ceramic composition [37,38]. Studies have demonstrated a significant decrease in cell viability for computer-assisted design (CAD) block feldspathic ceramics [47]. Feldspathic ceramic glass and glass-ceramics based on the 3CaO·P_2_O_5_-SiO_2_-MgO system were considered not cytotoxic by MTT ((3-(4,5-dimethylthiazol-2-yl)-2,5-diphenyltetrazolium bromide) [39] and neutral red analysis [48], respectively. Additionally, lithia–silica glass-ceramic was biocompatible, promoting cell adhesion and proliferation after 21 days [38]. However, zirconia-reinforced lithium silicate (ZLS) ceramic showed severe initial cytotoxicity when in contact with fibroblasts for 24 h [28]. This study demonstrated that fluorapatite glass-ceramic with or without SiC coating presented no cytotoxicity on human periodontal ligament cells by CellTiter-Blue. SiC-coated surfaces seeded with HPdLF cells may foster cell survival similar to surfaces that were not coated with SiC. Additionally, cells appeared elongated and intimately attached to both ceramic surfaces by SEM images. Some cell-to-cell interaction on the surface of ceramics was observed (Figure 8). Because the biocompatibility of the materials can be evaluated by the direct interaction between the ceramic surface and the cells, we can affirm that SiC coating is biocompatible. 

This study confirmed that SiC coating can decrease bacterial adhesion on the surface of ceramic veneers while maintaining biocompatibility with other cells. SiC recently demonstrated optimized color properties and corrosion resistance [21,22]. This coating can be applied to dental materials to improve their clinical performance.

## 5. Conclusions

A SiC coating on ceramic demonstrated enhanced bactericidal ability without inducing cytotoxic effects. The SiC coating presented bactericidal activity against *S. mutans* and *S. sanguinis* after 24 h of culture. In addition, the coating did not induce cytotoxic effects on periodontal ligament fibroblasts. These results indicate that a SiC coating can be used as a tool to prevent bacterial adhesion and improve the clinical performance of ceramic restorations. Further studies will include in vivo validation of coating performance. 

## Figures and Tables

**Figure 1 jfb-11-00033-f001:**
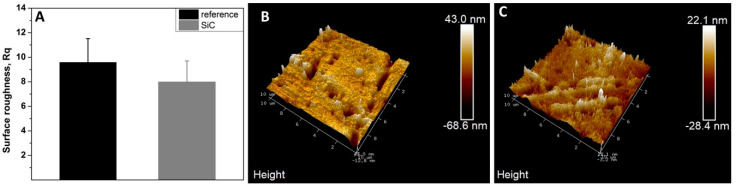
Roughness means values and standard deviation in non-coated (reference) and coated (ref SiC) samples (**A**). Atomic force microscopy (AFM) images of surface topography of non-coated (**B**) and coated SiC (**C**), with amplifications of 10 µm.

**Figure 2 jfb-11-00033-f002:**
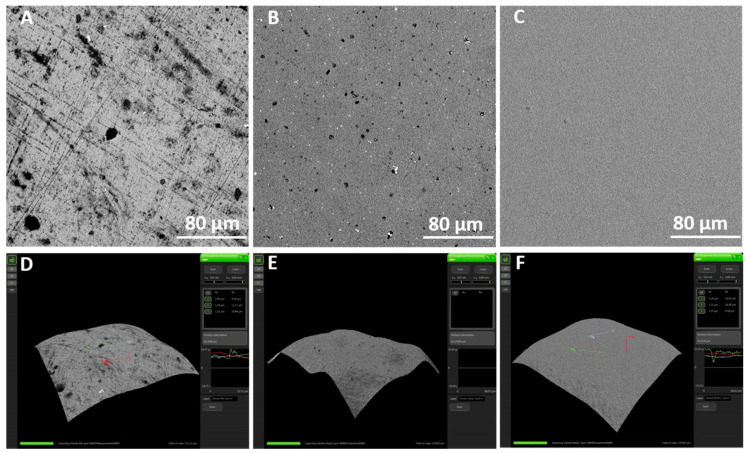
SEM images of roughness of non-coated and coated samples. (**A**,**D**) Non-coated fluorapatite disk, (**B**,**E**) glazed fluorapatite disk, and (**C**,**F**) SiC-coated fluorapatite disk.

**Figure 3 jfb-11-00033-f003:**
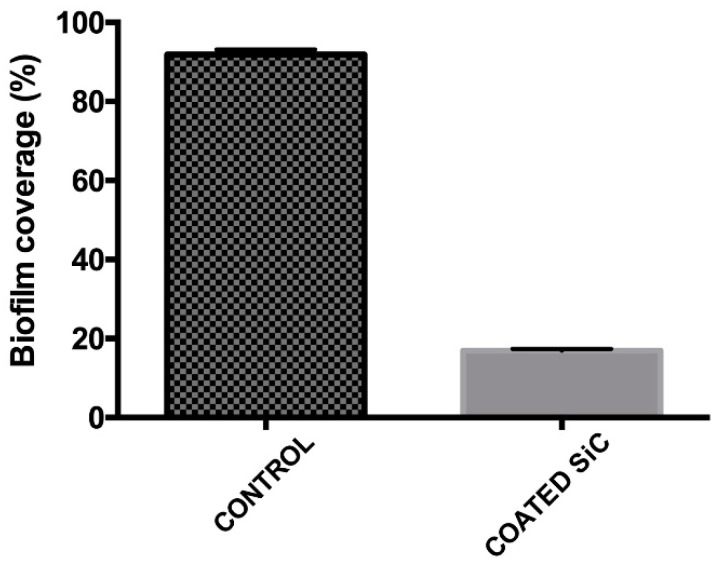
Live coverage of *Streptococcus mutans* and *Streptococcus sanguinis* after 24 h of culture on the non-coated (control) and coated surfaces.

**Figure 4 jfb-11-00033-f004:**
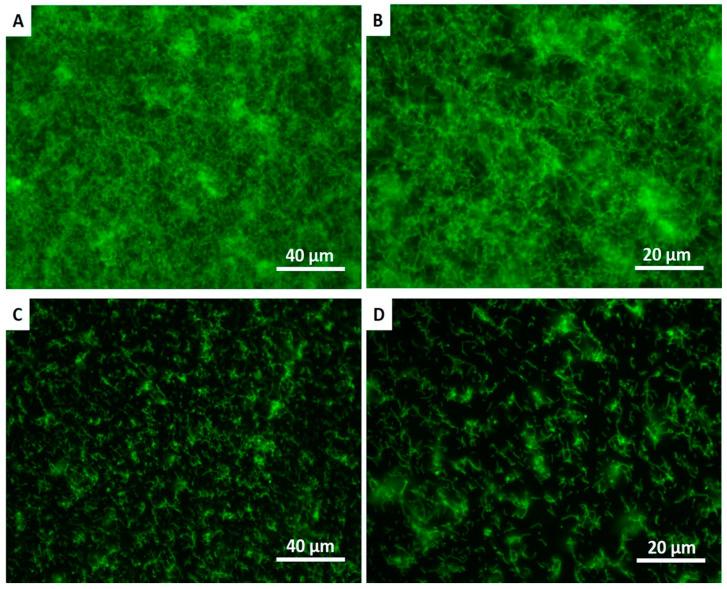
Live fluorescence images of *S. mutans* and *S. sanguinis* cultured for 24 h on the non-coated (**A**,**B**) and coated (**C**,**D**) surfaces. The cultures were stained with SYTO 9 to dye the living bacteria green.

**Figure 5 jfb-11-00033-f005:**
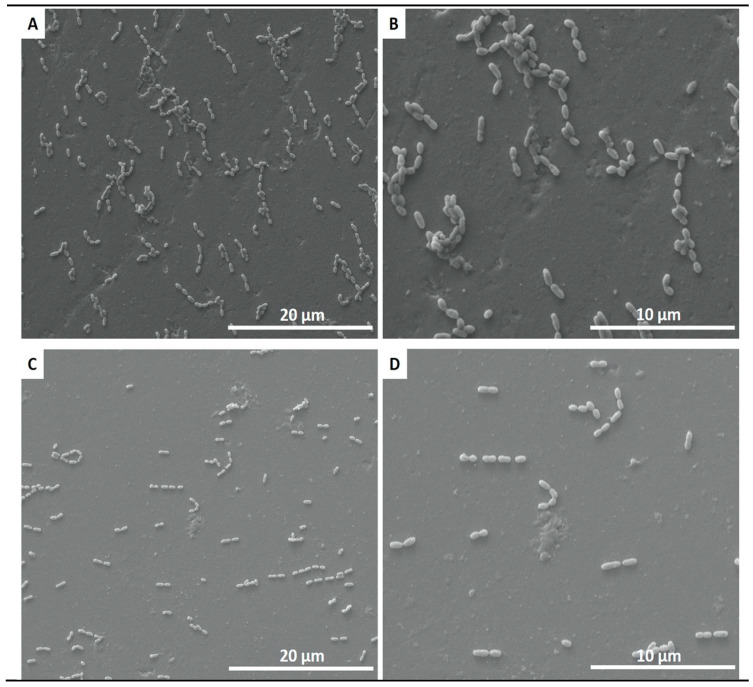
SEM adhesion of polymicrobial biofilm of *S. mutans* and *S. sanguinis* after 24 h of cultivation on non-coated (**A**,**B**) and coated (**C**,**D**) ceramic disks.

**Figure 6 jfb-11-00033-f006:**
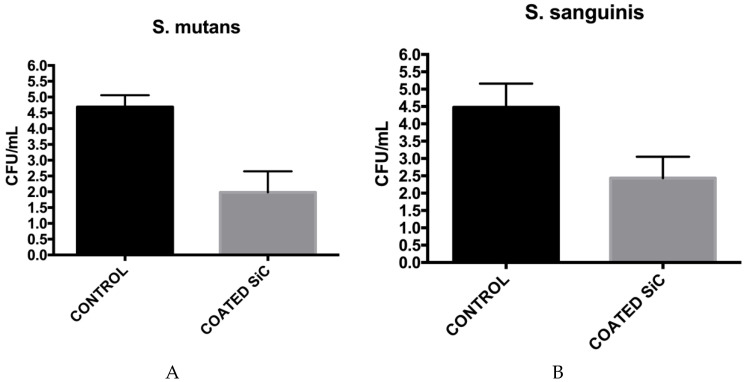
Mean (±SD) of colony-forming units (CFU)/mL of monomicrobial biofilms of *S. mutans* (**A**) and *S. sanguinis* (**B**) presented in the non-coated (control) and coated groups.

**Figure 7 jfb-11-00033-f007:**
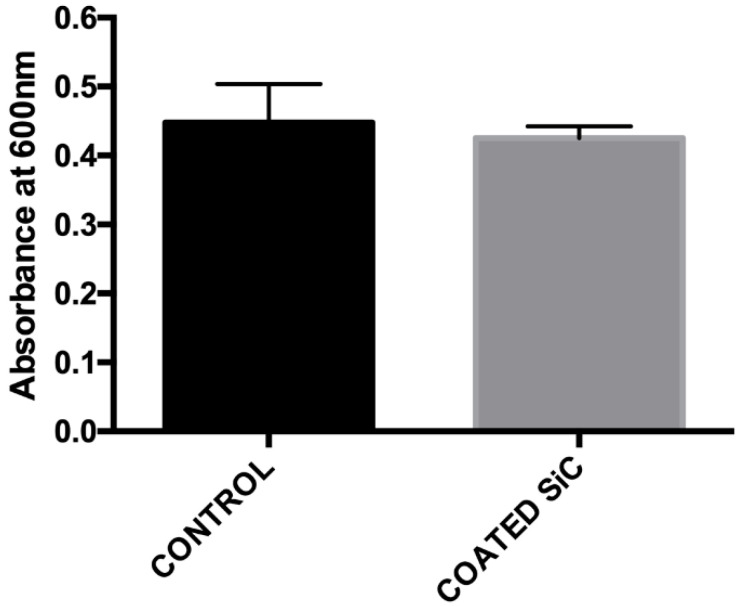
Cytotoxicity of the non-coated (control) and coated groups after 24 h of human periodontal ligament fibroblasts (HPdLF) culture assessed by CellTiter-Blue absorbance.

**Figure 8 jfb-11-00033-f008:**
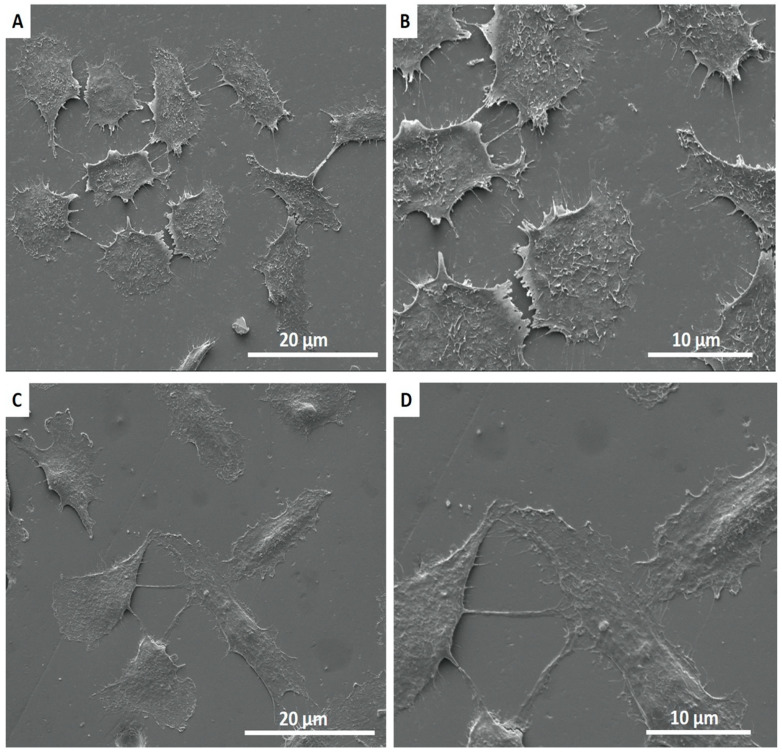
SEM images showing the adhesion of HPdLF on non-coated (**A**,**B**) and coated (**C**,**D**) surfaces after 24 h of incubation.

**Table 1 jfb-11-00033-t001:** Mean of contact angle measurements on non-coated and coated SiC surfaces.

Ceramic Samples	Contact Angle (±SD)
Non-coated	46° ± 2°
Coated SiC	60° ± 1°

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
