# Peer review of "Anti-Bacterial Properties and Biocompatibility of Novel SiC Coating for Dental Ceramic"

_jfb, 2020, doi:10.3390/jfb11020033_

Round 1

Reviewer 1 Report

This study is well managed, original and very intersting. It show with high performances that SiC coating can decrease bacterial adhesion on the surface of dental ceramic venners while maintaining bicompatibilty with other cells. I am agree with you for the need with further studies for SiC coating in vivo validation.

Just one suggestion: the explanation for the choice of the two bacterials tested is presented too far, in the discussion line 250-251. Could it be move in the 'material and methods' part?

Congratulation for your work.

Reviewer 2 Report

The manuscript presents a SiC coating on dental ceramic to improve anti-bacterial properties for the applications of dental prosthesis. Based on the comments below, I recommend the manuscript for publication with the Journal of Functional Biomaterials after major revision. The comments are as follows:

  1. The authors mentioned that in order to test cell proliferation on the SiC coating, the cytotoxicity was measured after 24 h. However, during 1 day of cell culture, cells adhered to the surface, with no proliferation. Therefore, please measure more time points for the interaction between cells and surfaces.
  2. In figure 1, the resolution is low. Please improve it.
  3. In figure 4, the scale bar is missed.
  4. In figure 5, please keep the size of SEM images the same and good image alignment.
  5. On page 8, line 217-219, you cannot claim any information about cell proliferation. Also, please give more description for Figure 8, e.g., cell morphology, cell and cell communications, etc.
  • Some noted errors to be corrected:
  1. Page 2, line 89, Is “(21)” a reference?
  2. Page 3, line 97 and 125, 24 h and 24 hours, please keep the units consistent, and the same everywhere else.
  3. Page 3, line 105, and page 4, line 154, (121oC, 60 min), 37oC.

Reviewer 3 Report

This paper deals with the study of the antibacterial properties of a SiC coating obtained by plasma chemical vapour deposition on dental ceramics. This work is properly addressed and adequately written. However, there are some points that must be improved:

  • The interest and the benefits of differentiating monomicrobial and polymicrobial adhesion must be explained.
  • Bacterial growth studies have been by comparison of uncoated and coated samples after 24 h. of polymicrobial formation. Thus, how has been monomicrobial activity tested?. This point must be clarified along the manuscript.
  • Different fluorescence and SEM images are shown only changing the scale. These are repetitive and do not provide further information (two differentiate scales are enough).
  • The weak point of this investigation is the poor discussion on the differences of adhesion results. Bacterial adhesion on surfaces is not a trivial issue and contradictory results of those reported here have been published in the literature along the last decades. Authors claim that hydrophobicity promotes microbial adherence, however, is widely known just the opposite: bacteria adhere better to hydrophobic surfaces.  Showed references at this point are related to specific ceramic surfaces. However result do not correspond to the general knowledge of bacteria adhesion to surfaces, therefore a more general discussion is  required.
  • Quantitative bacteria coverage of the surfaces is not addressed in the “results” section. How is it determined?

Round 2

Reviewer 2 Report

The revised manuscript can be accepted.

Reviewer 3 Report

Authors have addressed properly proposed recommendations.